# A Methionine-Portioning-Based Medical Nutrition Therapy with Relaxed Fruit and Vegetable Consumption in Patients with Pyridoxine-Nonresponsive Cystathionine-β-Synthase Deficiency

**DOI:** 10.3390/nu15143105

**Published:** 2023-07-11

**Authors:** Esma Uygur, Cigdem Aktuglu-Zeybek, Mirsaid Aghalarov, Mehmet Serif Cansever, Ertugrul Kıykım, Tanyel Zubarioglu

**Affiliations:** 1Department of Pediatric Nutrition and Metabolism, Cerrahpasa Faculty of Medicine, Istanbul University-Cerrahpasa, 34098 Istanbul, Turkey; dytesmauygur@gmail.com (E.U.); dracaz@iuc.edu.tr (C.A.-Z.); ertugrul.kiykim@iuc.edu.tr (E.K.); 2Nutrition and Dietetics PhD Programme, Institute of Health Sciences, Acibadem Mehmet Ali Aydınlar University, 34752 Istanbul, Turkey; 3Department of Pediatrics, Cerrahpasa Faculty of Medicine, Istanbul University-Cerrahpasa, 34098 Istanbul, Turkey; mirseidagalarov94@gmail.com; 4Division of Medical Laboratory Techniques, Department of Medical Documentation and Techniques, The Vocational School of Health Services, Istanbul University-Cerrahpasa, 34295 Istanbul, Turkey; mehmet.cansever@iuc.edu.tr

**Keywords:** CBS deficiency, homocystinuria, homocysteine, simplified diet, methionine portioning, fruits, vegetables

## Abstract

The main treatment for pyridoxine-nonresponsive cystathionine-β-synthase deficiency is a strict diet. Most centers prescribe low-protein diets based on gram–protein exchanges, and all protein sources are weighed. The purpose of this study is to investigate the effects of a more liberal methionine (Met)-based diet with relaxed consumption of fruits and vegetables on metabolic outcomes and dietary adherence. Ten patients previously on a low-protein diet based on a gram–protein exchange list were enrolled. The natural protein exchange lists were switched to a “Met portion exchange list”. Foods containing less than 0.005 g methionine per 100 g of the food were accepted as exchange-free foods. The switch to Met portioning had no adverse effects on the control of plasma homocysteine levels in terms of metabolic outcomes. It resulted in a significant reduction in patients’ daily betaine dose. All patients preferred to continue with this modality. In conclusion, methionine-portion-based medical nutrition therapy with relaxed consumption of fruits and vegetables seems to be a good and safe option to achieve good metabolic outcomes and high treatment adherence.

## 1. Introduction

Cystathionine-β-synthase (CBS) deficiency, also known as classical homocystinuria, is an autosomal recessively inherited metabolic disorder that results in accumulation of homocysteine (Hcy) and methionine (Met) but, contrarily, depletion of cystathionine and cysteine due to mutations in the *CBS* gene. The clinical phenotype can vary widely, ranging from severe multisystemic disease in early childhood to asymptomatic patients diagnosed in adulthood. The major clinical findings include neuropsychiatric signs, such as autism, mental retardation and psychosis, skeletal system malformations, including osteoporosis, predisposition to thromboembolism, and ocular lens dislocation [1,2,3].

The defective enzymatic activity of CBS results in alteration in transsulfuration and enhanced remethylation, so the main biochemical features are increased plasma total homocysteine (tHcy) and Met concentrations. Measurement of increased tHcy and Met concentrations and decreased enzyme activity in fibroblasts and plasma can be supportive, and a definite diagnosis can be conducted by detecting biallelic pathogenic variants in the *CBS* gene [4,5,6,7]. Early treatment can lead to a favorable metabolic outcome, and inclusion of CBS deficiency in neonatal screening programs is strongly recommended for a good clinical prognosis [8,9]. In the case of diagnostic delay and inadequate metabolic control, morbidity and mortality may occur due to multisystemic involvement of the central nervous system, vascular system, and connective tissue [4].

Homocysteine-induced oxidative stress, endoplasmic reticulum stress with apoptosis of endothelial cells, induction of unfolded protein response, and chronic inflammation are the potential underlying mechanisms responsible for clinical symptoms [10,11]. Therefore, the main goal of treatment is to lower or normalize plasma homocysteine concentrations, and treatment should be continued throughout life. Cystathionine-β-synthase deficiency is divided into subgroups depending on the response of patients to pyridoxine. In patients who do not respond to pyridoxine, the main therapeutic strategy is protein-restricted medical nutrition therapy used alone or in combination with betaine [4]. Inborn errors of metabolism (IEM) centers use different protocols for medical nutrition therapy. As there is not a uniform consensus, both low-Met and low-natural protein diets have been used in conjunction with a methionine-free l-amino acid mixture (MFAAM) in patients with CBS deficiency. However, dietary adherence is often poor and can be difficult, especially in late-diagnosed patients with mental involvement [12,13]. Betaine is preferred as an add-on therapy in patients whose plasma tHcy levels cannot be maintained within the targeted ranges with medical nutrition therapy. However, it has been reported that betaine intake is associated with various side effects in some patients due to high plasma Met levels [14]. In conclusion, individualized precision medicine according to patients’ clinical and biochemical parameters is recommended instead of a standardized therapeutic regimen to achieve a good metabolic outcome.

In the medical literature, liberalization of the diet according to the phenylalanine (Phe) content of various foods and unrestricted consumption of fruits and vegetables has been described as a successful treatment option for metabolic control in patients with phenylketonuria (PKU) [15,16,17,18,19]. Inspired by the positive results of PKU studies, in this study, we aimed to investigate the effects of Met-portioning-based medical nutrition therapy with freer consumption of fruits and vegetables compared with strict protein change diets regarding metabolic outcome and dietary adherence.

## 2. Materials and Methods

### 2.1. Study Design and Participants

This prospective clinical trial was conducted between May 2020 and April 2021 at Istanbul University Cerrahpasa, Cerrahpasa Medical Faculty, Department of Pediatric Nutrition and Metabolic Diseases with pyridoxine-nonresponsive CBS deficiency patients. The diagnosis of CBS deficiency was defined as a pathogenic biallelic mutation in the *CBS* gene in patients with positive clinical and biochemical findings consistent with the disease.

Patients were defined as nonresponsive if tHcy fell less than 20%, partially responsive if it fell more than 20% but remained above 50 μmol/L, and fully responsive if tHcy fell below 50 μmol/L [4].

Patients diagnosed with pyridoxine-nonresponsive CBS deficiency were included if they were receiving protein-restricted medical nutrition therapy based on a gram–protein calculation, monitored regularly, and plasma tHcy samples were collected at the recommended frequency. Patients who did not have molecular confirmation of the diagnosis, who were diagnosed with pyridoxine-responsive or partially responsive CBS deficiency, and who did not receive regular clinical and laboratory monitoring were excluded from the study.

All procedures used were in accordance with the ethical standards of the local ethics committee of Cerrahpasa Medical Faculty (07/04/2021-69864) and the Helsinki Declaration of 1975, as revised in 2013. All parents of patients included in this study provided informed consent.

### 2.2. Details of Dietary Interventions and Switching Process to Methionine Portioning

The study period was divided into two periods. In period 1 (May 2020–October 2020), medical nutrition therapy consisted of MFAAM and natural protein arranged according to a “gram protein exchange list”. All foods were weighed. In period 2 (November 2020–April 2021), patients continued to receive MFAAM, but natural protein exchange lists were switched according to a “Met portioning exchange list”. Since an exchange system based on 25 mg is used in our center for all amino acid metabolism disorders (e.g., in maple syrup urine disease 25 mg leucine, in PKU 25 mg Phe), 25 mg methionine was considered as one portion for these patients. Foods that contained less than 0.005 g of methionine per 100 g of the food were accepted as exchange-free foods and were allowed to be consumed liberally without calculating the Met content. The other foods were weighed by the patients according to the Met portioning list. The methionine and protein exchange lists were calculated using the United States Department of Agriculture (USDA) database [20]. A comparison of the amounts of some fruits and vegetables according to the Met portioning list and gram–protein exchange list is shown in Table 1. Table 2 provides a list of the exchange-free foods and a comparison of their amounts according to the Met portioning list and the gram–protein exchange list.

At the time of switching to period 2, patients and their caregivers were called to the hospital and nutrition education was provided. For those individuals who were unable to attend the face-to-face meetings, online education was provided via WhatsApp or Zoom. After the patient’s education was completed, medical nutrition therapy based on gram–protein calculation was switched to medical nutrition therapy based on methionine portioning. At switching, detailed 3-day food consumption records were kept, the amount was calculated and converted to portions, and patients’ diets were revised according to the Met exchange list.

In both periods, patients’ medical nutrition therapy was arranged according to the plasma tHcy levels, patients’ age, body weight, and food records. The principles of follow-up, including the target plasma tHcy levels and the recommended frequency of sampling, were not changed during the two periods. According to the treatment protocol for patients with CBS deficiency, we administered vitamin B6 (10 mg/kg/day, maximum 500 mg/day) and folic acid (5 mg/day) to each patient, even if they did not respond to B6 in either period. Patients were also closely monitored for vitamin B12 deficiency. The dosage of the vitamins was not changed during the study.

In this study, the clinical and biochemical data of the patients were reviewed and the following items were recorded for both periods: current age, age at diagnosis, sex, phenotypic characteristics, *CBS* gene molecular analysis, anthropometric characteristics (height, body weight, and body mass index), medical and nutritional treatments they received. The energy, total protein, and methionine amounts of the dietary therapy in both periods, dietary adherence, total energy, protein, and methionine intakes of the 3-day food consumption records were recorded and analyzed using the Nutrition Information System (BeBIS 8.0). Plasma tHcy levels, dose of betaine treatment, and percentage of achievement of good metabolic control according to the target ranges for tHcy in the CBS deficiency guideline (4) were compared between the two periods.

### 2.3. Assessment of Dietary Adherence

In period 1, dietary adherence was assessed mainly by 3-day food consumption records and patient visits. Patients’ detailed food consumption records observed the foods they consumed both at school/work and at home. In addition, adherence to protein restriction and consumption of MFAAM were identified. At the end of period 2, a questionnaire was administered on the effects of the new dietary approach. A total of six questions (multiple choice or short answer) were asked:Did the new Met exchange system provide ease of application according to the diet you applied before?Did it provide you convenience in terms of food variety?Has the frequency of consumption of prohibited foods decreased?Did the new exchange system cause more fruit and vegetable consumption?Did the new exchange system improve your dietary adherence at school/work?Would you like to continue with the new exchange system?

### 2.4. Statistical Analysis

Statistical analyses were performed using Statistical Package for Social Sciences version 26.0 (SPSS, Inc., Chicago, IL, USA). Continuous variables are presented as median (minimum or maximum) or mean ± standard deviation (SD), and categorical variables as numbers or percentages. Normal distribution of the data was evaluated with a Kolmogorov–Smirnov test. The analysis of normally distributed continuous variables in two dependent pairs was performed with the paired samples *t*-test, while the Mc-Nemar test was used for the analysis of categorical variables used for comparison of groups where appropriate. A value of *p* < 0.05 was considered statistically significant.

## 3. Results

Ten patients were included in the study. Five patients (50%) were female and five (50%) were male. The mean age of the patients was 15.8 ± 9.29 years, and the mean follow-up time after the diagnosis of CBS deficiency was 9.25 ± 7.33 years. Nine patients (90%) exhibited a clinical feature relevant to CBS deficiency. The only asymptomatic patient was diagnosed by newborn screening after the diagnosis of CBS deficiency in the first sibling of the family. Patient demographics and characteristics are shown in Table 3.

### 3.1. Comparison of Anthropometric Measurements and Components of Medical Nutrition Therapy

Switching to the Met portion exchange list did not result in a statistically significant difference in patients’ height z-score, body weight z-score, and body mass index (*p* = 0.085, *p* = 0.522, and *p* = 0.242, respectively). In period 1, all patients received a low-protein diet in combination with MFAAM. MFAAM amount and natural protein intake were not statistically different in period 2 (*p* = 0.873 and *p* = 0.284, respectively). Data on anthropometric measurements and medical nutrition therapy components are shown in Table 4 and Table 5.

### 3.2. Assessment of Metabolic Outcome after Switching to the Met Portioning Exchange List

In period 1, plasma tHcy levels below 100 μmol/L were measured in all patients on betaine and medical nutrition therapy. Since switching to the Met portion exchange list did not result in a statistically significant difference in patients’ plasma tHcy levels (*p* = 0.250), dietary relaxation in period 2 was not considered to have a negative effect on metabolic outcome. Moreover, the daily betaine dose of patients was significantly reduced in the second period (*p* = 0.017). Betaine treatment was discontinued in four patients (40%) because metabolic control could be achieved by diet alone. Data comparing plasma tHcy levels and betaine doses in both periods are shown in Table 5. The change in plasma Met levels as a result of dietary modification and withdrawal or reduction in betaine is shown in Table 6.

### 3.3. Efficacy of Met Portioning Exchange List on Dietary Adherence

Nine patients participated in the survey. The adult patients (P1, P9, P10) and the 16-year-old patient (P8) answered the questionnaires themselves; the questionnaires of the other patients were answered by their caregivers. Because patients 3, 4, and 5 were sibling pairs from the same family, the same parent answered the questions for these three patients. According to the results of the survey, all the patients indicated that this new dietary approach was easier to implement, increased the variety of foods, and helped to prevent the consumption of prohibited foods. Six patients (66.7%) reported consumption of more fruits and vegetables, and seven patients (77.7%) reported better adherence to the diet in the second period at school or work. All nine patients planned to continue with the new dietary approach.

## 4. Discussion

In the literature, liberalization of the diet according to the phenylalanine content of various foods and unrestricted consumption of fruits and vegetables has been described as a successful treatment option for metabolic control in PKU patients. Inspired by the positive results of PKU studies, we aimed to use a similar model based on methionine portioning in this study. A simplified and feasible diet, mainly rearranged according to Met portioning exchange lists and allowing unrestricted intake of fruits and vegetables containing less than 5 mg/100 g of Met, was used to treat pyridoxine-nonresponsive CBS deficiency patients who were previously receiving low-protein diets based on gram–protein exchange lists in which all foods were weighed. We demonstrated that Met-portioning-based medical nutrition therapy and nonweighted inclusion of all fruits and vegetables in the exchange-free food list had no adverse effects on the control of plasma tHcy levels with respect to metabolic outcome. Switching to Met portioning resulted in a significant reduction in patients’ daily betaine dose, and 40% of the patients discontinued betaine because metabolic control could be achieved with dietary therapy alone. The results of an informal, nonstandardized questionnaire indicated that patients felt adherence to therapy may be facilitated, and that patients preferred to continue with the new modality. To the best of our knowledge, this study is the first to demonstrate diet liberalization by unrestricted consumption of fruits and vegetables with a Met content of no more than 5 mg/100 g, and, in particular, this simplified dietary model did not affect long-term control of blood Hcy levels.

The main goal of treatment is to normalize or reduce plasma tHcy levels in patients with CBS deficiency; in individuals who do not respond to pyridoxine, treatment consists mainly of medical nutrition therapy and betaine [4]. To date, dietary approaches to CBS deficiency have been studied in the medical literature. A large-sample-size study involving 29 IEM centers and 181 patients with CBS deficiency compared current dietary practices. Fifteen of the twenty-nine centers prescribed a diet based primarily on Met analysis of foods. However, almost all these centers used a combination of Met and natural protein analysis, including grams of protein exchanges, or calculated the protein content of all foods consumed. There was no free food list, and metabolic outcome data were not available [12]. The results of our study suggest that metabolic control can be achieved by using a Met portioning list as a natural protein source. In addition, liberalization of the diet by a free food list appeared to be safe and had no negative effects on metabolic control.

Another point of controversy in treatment is the use of betaine in CBS deficiency. Most IEM centers use betaine as an add-on treatment to lower plasma tHcy levels to the target range as recent studies have provided evidence for the clinical safety of betaine. A study of 130 individuals found that betaine use was associated with diarrhea and visual disturbances, but these resolved without sequelae. Plasma Met levels exceeded 1000 μmol/L in only one patient, but he had no signs of cerebral edema [21]. In another study, 277 adverse effects were reported by patients with different etiologies, leading to homocystinuria. Only three adverse effects were associated with betaine. Two of them suffered from bad taste and headache, and one from interstitial lung disease. However, the patient who developed interstitial lung disease also had cobalamin C deficiency, and these respiratory problems may also have been due to the underlying disease [22]. Despite these studies supporting the safety of betaine, several patients have been reported to develop hypermethioninemia greater than 1000 μmol/L, and clinical findings consistent with increased intracranial pressure and cerebral edema on magnetic resonance imaging after the fourth week of betaine administration have been reported [23,24,25]. In addition to safety concerns, murine studies have shown that medical nutrition therapy is superior to betaine in terms of metabolic control, and the efficacy of betaine in lowering plasma tHcy levels has been shown to decrease over time with long-term betaine use [26,27]. Reducing the daily betaine dose was not a primary objective in our study. However, according to our results, metabolic control can be achieved in four patients by dietary treatment alone, and betaine treatment was discontinued by switching to Met portioning. Continuation of treatment as monotherapy was evaluated in favor of patients in terms of both convenience and safety.

Cystathionine-β-synthase deficiency has been reported to be challenging for patients and caregivers in terms of dietary adherence and quality of life. Because CBS deficiency is not covered by an expanded newborn screening program in most countries, patients are usually diagnosed late, and patients who did not have dietary restrictions until diagnosis have difficulty adhering to treatment with Met and a natural low-protein diet, leading to quality of life problems [28,29]. Irregular or incomplete intake of MFAAM, complexity of dietary regimens, the need to weigh all foods before consumption, and socioeconomic factors, including family characteristics, family education level, and patient sense of responsibility, have been described as factors complicating dietary adherence in patients with CBS deficiency [13,30]. In a study that addressed dietary adherence problems in patients with CBS deficiency, it was found that 81% of patients had difficulty adopting a low-protein diet. In addition, 73% of patients reported that they had difficulty weighing foods [13]. In our study, the increase in the variety of foods that patients could consume without weighing led to a simplification of their diet in terms of its impact on their school, work, and social lives.

Our study had some limitations. The first important limitation was the small sample size of our study. The study was conducted with a limited number of patients, mainly young patients. The main reason was that even patients who responded partially to pyridoxine might bias the decision on the efficacy of dietary change, so only patients who did not respond to pyridoxine were included in our study. The second important limitation was the informal, nonstandardized nature of the questionnaire rather than a more neutral questionnaire—for example, an interval scale—that might prevent facilitating responses in favor of period 2. Another limitation was that only patients who adhered to regular follow-up were included and the data were collected during a relatively short follow-up period.

## 5. Conclusions

Our data suggest that methionine-portioning-based medical nutrition therapy with relaxed fruit and vegetable consumption seems to be a good and safe option to maintain good metabolic outcomes and dietary adherence.

## Figures and Tables

**Table 1 nutrients-15-03105-t001:** Comparison of the amounts of some fruits and vegetables according to the methionine portioning and the gram–protein exchange lists [20].

Fruits and Vegetables	1 Met Portion	1 Protein Portion
Broccoli	65 g	35 g
Avocado	67 g	51 g
Potato	78 g	48 g
Brussels Sprouts	78 g	29 g
Mushrooms	80 g	32 g
Kale	86 g	34 g
Eggplant	227 g	102 g
Banana	312 g	91 g
Cucumber	416 g	153 g
Tomato	416 g	102 g
Watermelon	416 g	163 g

Met: methionine.

**Table 2 nutrients-15-03105-t002:** List of exchange-free foods and comparison of their amounts according to the methionine portioning and gram–protein exchange lists [20].

Fruits and Vegetables	1 Met Portion	1 Protein Portion
Melon	500 g	185 g
Celery	500 g	145 g
Lettuce	500 g	111 g
Grapefruit	500 g	181 g
Nectarine	500 g	94 g
Loquat	625 g	232 g
Gourd	625 g	161 g
Cranberry	833 g	217 g
Pear	1250 g	277 g
Strawberry	1250 g	149 g
Lime	1250 g	142 g
Tangerine	1250 g	123 g
Papaya	1250 g	212 g
Onion	1250 g	90 g
Apple	2500 g	370 g

Met: methionine.

**Table 3 nutrients-15-03105-t003:** Demographic data and characteristics of patients.

	Sex	Age at Diagnosis (months)	Current Age (years)	Consanguinity	Mutation	Protein Change	Clinical Phenotype
P1	Female	60	20	(+)	c.919G>A	p.G3075	thromboembolic episodes mental retardation lens dislocation
P2	Female	3	7	(+)	c.919G>A	p.G3075	asymptomatic
P3	Female	76	10	(+)	c.1240G>T	p.V414F	lens dislocation
P4	Male	65	9	(+)	c.1240G>T	p.V414F	thromboembolic episodes mental retardation lens dislocation
P5	Male	65	9	(+)	c.1240G>T	p.V414F	lens dislocation
P6	Male	123	12	Ø	c.752T>C c.1064 C>T	p.L251P p.A355V	thromboembolic episodes mental retardation lens dislocation skeletal abnormalities
P7	Female	72	13	(+)	c.775G>A	pG259S	lens dislocation
P8	Female	90	16	Ø	c.752T>C c.1152G>C	p.L251P p.K384N	thromboembolic episodes mental retardation lens dislocation
P9	Male	156	37	(+)	c.1152 G>C	p.K384N	mental retardation lens dislocation skeletal abnormalities
P10	Male	84	25	(+)	c.1058C>T	p.Thr353Met	mental retardation lens dislocation skeletal abnormalities

P1 and P2, P3, P4, and P5 were sibling pairs from the same family.

**Table 4 nutrients-15-03105-t004:** Comparison of anthropometric measurements between medical nutrition therapy based on gram–protein portioning and methionine portioning.

	Period 1 g Protein Exchange List	Period 2 Met Portioning Exchange List	*p*-Value
Height *z*-score *	0.58 ± 1.63	1.04 ± 1.99	0.087
Weight *z*-score	0.00 ± 1.24	0.08 ± 1.39	0.522
BMI (kg/m^2^)	17.67 ± 2.73	18 ± 2.71	0.242

* Analysis of *z*-scores for height was performed in pediatric patients and adult patients were excluded. Data were presented as mean ± SD. Continuous data were analyzed by the paired sample *t*-test. Met: methionine; BMI: body mass index.

**Table 5 nutrients-15-03105-t005:** Comparison of dietary components, betaine consumption, and plasma total homocysteine and methionine levels between medical nutritional therapies based on gram–protein portioning and methionine portioning.

	Period 1 g Protein Exchange List	Period 2 Met Portioning Exchange List	*p*-Value
Dietary natural protein (g)	21.8 ± 10.46	25.15 ± 8.36	0.284
MFAAM (g/kg)	1.14 ± 0.61	1.13 ± 0.54	0.873
Plasma tHcy (µmol/L)	61.83 ± 17.21	52.62 ± 25.54	0.250
Betaine dose (g/day)	4.5 ± 1.58	2.3 ± 2.35	**0.017**
Plasma Met (µmol/L)	285.61 ± 272.604	203.54 ± 161.66	0.369

Data were presented as mean ± SD. Continuous data were analyzed by the paired sample *t*-test. Bold values indicate statistically significant *p*-values (*p* < 0.05). Met: methionine; MFAAM: methionine-free L-amino acid supplement; tHcy: total homocysteine; SD: standard deviation.

**Table 6 nutrients-15-03105-t006:** Data on daily betaine dose, plasma methionine, and homocysteine levels in both periods.

	Period 1 g Protein Exchange List			Period 2 Met Portioning Exchange List		
	Plasma Met (µmol/L)	Betaine Dose (g/day)	Plasma tHcy (µmol/L, mean ± SD)	Plasma Met (µmol/L)	Betaine Dose (g/day)	Plasma tHcy (µmol/L, mean ± SD)
P1	572	6	79.37 ± 21.64	96	3	52.02 ± 1.5
P2	576	6	43.24 ± 28.1	240	Ø	37.58 ± 15.15
P3	29	3	50.26 ± 14.11	71	Ø	59.53 ± 19.94
P4	32	3	47.23 ± 8.47	66	Ø	37.63 ± 15.78
P5	40	3	48.13 ± 10.41	79	Ø	32.03 ± 15.65
P6	84	3	75.62 ± 55.16	126	2	30.23 ± 28.07
P7	167	6	87.5 ± 16.2	420	6	72.21 ± 9.59
P8	109	3	69.25 ± 16.61	329	6	113.66 ± 23.15
P9	627	6	42.15 ± 1.20	102	3	34.64 ± 12.22
P10	618	6	75.55 ± 17.46	506	3	56.69 ± 24.62

Met: methionine; tHcy: total homocysteine; SD: standard deviation.

## Data Availability

The data that support the findings of this study are available from the corresponding author, T.Z., upon reasonable request.

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
