# Peer review of "A Methionine-Portioning-Based Medical Nutrition Therapy with Relaxed Fruit and Vegetable Consumption in Patients with Pyridoxine-Nonresponsive Cystathionine-β-Synthase Deficiency"

_nutrients, 2023, doi:10.3390/nu15143105_

Round 1
Reviewer 1 Report
This paper tried to propose a more flexible diet for the patients with pyridoxine nonresponsive cystathionine-β-synthase deficiency, it is quite helpful to improve the life quality of those patients and it may serve as an add-on therapy as well. My only concern is that the sample volume is too small, and the age of the patients is too young to expand the findings to all the patients. Although the authors mentioned that in the discussion section, but they did not discuss it in detail, nor did they do anything to improve it. I could not recommend its publication until it is properly addressed.
Reviewer 2 Report
Review
This work aims at liberalising the diet for patients with classical homocystinuria by changing the approach from protein exchanges to Met exchanges.The general idea has already been established in Phenylketonuria, and it is intriguing to investigate this in CBS deficiency.
My comments focus on the methods and the clarity of the scientific content.
Line 91: QoL is not measured here.
Line 94: please mention as a limitation that including only patients adhering to regular follow-up means a selection bias. Furthermore, the absence of a control group that has only received the intense educational intervention on food and diet is a main limitation of this work.
Line 105: 1 portion of Met is 25 mg – please clarify: is that for practical reasons or where do those 25 originate from?
Line 111: for the interest of the reader – do other databases have comparable estimates of Met for the food groups or is there a huge difference between the USDA and other databases?
Lines 140: I do not completely follow how the switch was done. Did you estimate protein intake from the 3-day food consumption records to estimate / calculate Met intake, and then use this Met intake as a start? Please clarify.
How did you estimate “compliance and adherence” (please make it adherence only) in period 1? I understand that you have applied only one questionnaire after period 2.
Line 148: & table 6: The questionnaire is constructed in a way that results in extreme bias. All items have a “direction” and facilitate answers in favour of period 2. To assess adherence, data should have been gathered using an, at best, standardised, but at least, a neutral questionnaire that could have been applied after each period. I am afraid that the data as presented do not at all reflect adherence and the data and related conclusions should be omitted.
A minor comment to Q 5: how can this question (‘how did it change …’) be answered with YES/NO?
Who did answer the questions: how many patients and how many caregivers? It is well known that self- and proxy reporting yield different results.
Line 181: are there any pairs of siblings in the sample?
Table 4: Height – 3 patients are adults and will not grow between period 1 and 2. Analysis should only include patients with the potential of change.
Table 5: Methionine data should be presented, best in a figure, to show effects of change of diet and betaine withdrawal / reduction. P-values: unclear, to which parameters they relate: you are showing single cases, but p-values are for the analyses in the whole group, please clarify.
Line 213: text & figure 1: please do not show non-significant data and please do not introduce an arbitrarily chosen cut-off of 75 for tHcy.
Line 278: here is a new goal / aim of the study – to reduce betaine. If it was an aim of this study from the beginning, please clarify earlier.
Minor, typos:
Line 45 defective.. deficient – please correct
Line 280: please rephrase “safe spectrum”
Round 2
Reviewer 1 Report
Since my concerns have been properly addressed, I recommend its publication in the journal of Nutrients.
Author Response
The authors thank the reviewer for considering the text.
Reviewer 2 Report
The authors have achieved an improvement of their work, but there are still some open questions.
Mainly, I do not agree that results from a non-standardized, and, even more limiting, unbalanced questionnaire should be included because the authors find that “the removal of all data and conclusions” … related to it… “could cause a large gap in the text”.
Therefore, I suggest to
- Remove sentence in line 26, and first half of sentence in line 29 from the abstract.
- Omit table 7
- Delete sentence in line 238 referring to table 7
- Rephrase line 264 to: “The results of an informal, non-standardised questionnaire indicated that patients felt adherence to therapy may be facilitated, and that patients preferred to continue with the new modality.”
- Delete in line 265 “that this new nutritional approach significantly increased patients' adherence to therapy”
- Delete sentence in line 280, it is not nine patients participating but what you explain in the following sentence.
- Clarify how many siblings were represented by one couple of parents / one parent filling your questionnaire – this may represent an additional bias if the number of independent answers is even lower.
- Delete in line 322 “This resulted in 77.7% of patients better adhering to their diet.”
- Line 328: an interval scale used for pre- and post testing could have easily been constructed for your study. Thus, please rephrase “The second important limitation was the lack of a more neutral questionnaire-for example, an interval scale-that might prevent facilitating responses in favour of period 2.”
Other issues:
I suggest adding methionine levels (means, range) also to table 5.
Table 6 shows that P3, 4, 5 had extremely low Met – sure, that they were not B6 responsive? Were they treated with B6? Please add B6 treatment data for all patients and add individual tHcy concentrations to table 6.
Are patients 3,4,5 triplets? Only this would explain that they were all diagnosed at the same age. Please clarify.
Minor:
Line 161, rephrase, since adherence is mentioned three times (adherence to protein restriction, consumption of prohibited foods, dietary adherence): “In addition, adherence to protein restriction, consumption of MFAAM, and consumption of prohibited foods were identified and dietary adherence was assessed”
Line 280: rephrase “Compared with our study, the results of our study…”
What is meant by “patient ownership”, line 315?
Line 316: “In a study that addressed dietary adherence problems in patients with CBS deficiency, consumption of MFAAM was considered a problem in 74% of patients.”
True, but the new dietary approach did not result in reduction of amino acids. Suggest deleting.
The authors have achieved an improvement of their work, but there are still some open questions.
Mainly, I do not agree that results from a non-standardized, and, even more limiting, unbalanced questionnaire should be included because the authors find that “the removal of all data and conclusions” … related to it… “could cause a large gap in the text”.
Therefore, I suggest to
- Remove sentence in line 26, and first half of sentence in line 29 from the abstract.
- Omit table 7
- Delete sentence in line 238 referring to table 7
- Rephrase line 264 to: “The results of an informal, non-standardised questionnaire indicated that patients felt adherence to therapy may be facilitated, and that patients preferred to continue with the new modality.”
- Delete in line 265 “that this new nutritional approach significantly increased patients' adherence to therapy”
- Delete sentence in line 280, it is not nine patients participating but what you explain in the following sentence.
- Clarify how many siblings were represented by one couple of parents / one parent filling your questionnaire – this may represent an additional bias if the number of independent answers is even lower.
- Delete in line 322 “This resulted in 77.7% of patients better adhering to their diet.”
- Line 328: an interval scale used for pre- and post testing could have easily been constructed for your study. Thus, please rephrase “The second important limitation was the lack of a more neutral questionnaire-for example, an interval scale-that might prevent facilitating responses in favour of period 2.”
Other issues:
I suggest adding methionine levels (means, range) also to table 5.
Table 6 shows that P3, 4, 5 had extremely low Met – sure, that they were not B6 responsive? Were they treated with B6? Please add B6 treatment data for all patients and add individual tHcy concentrations to table 6.
Are patients 3,4,5 triplets? Only this would explain that they were all diagnosed at the same age. Please clarify.
Minor:
Line 161, rephrase, since adherence is mentioned three times (adherence to protein restriction, consumption of prohibited foods, dietary adherence): “In addition, adherence to protein restriction, consumption of MFAAM, and consumption of prohibited foods were identified and dietary adherence was assessed”
Line 280: rephrase “Compared with our study, the results of our study…”
What is meant by “patient ownership”, line 315?
Line 316: “In a study that addressed dietary adherence problems in patients with CBS deficiency, consumption of MFAAM was considered a problem in 74% of patients.”
True, but the new dietary approach did not result in reduction of amino acids. Suggest deleting.
